# Assessment of Personal Occupational Exposure to Radiofrequency Electromagnetic Fields in Libraries and Media Libraries, Using Calibrated On-Body Exposimeters

**DOI:** 10.3390/ijerph16122087

**Published:** 2019-06-13

**Authors:** Amelie Massardier-Pilonchery, Elena Nerrière, Sophie Croidieu, Fabien Ndagijimana, François Gaudaire, Christophe Martinsons, Nicolas Noé, Martine Hours

**Affiliations:** 1Université de Lyon, Université Claude Bernard Lyon1, Ifsttar, UMRESTTE, UMR T_9405, 8 avenue Rockefeller 69373 Lyon, France; e_nerriere@yahoo.fr (E.N.); martine.hours@ifsttar.fr (M.H.); 2Centre Hospitalier Lyon Sud, Hospices Civils de Lyon, 69495 Pierre Bénite, France; 3Service de médecine préventive de la Ville Lyon, 69006 Lyon, France; sophie.croidieu@mairie-lyon.fr; 4Université Grenoble Alpes, 38400 Saint Martin d’Hères, France; fabien.ndagijimana@univ-grenoble-alpes.fr; 5Division Eclairage et Champs Electromagnétiques, Centre Scientifique et Technique du Bâtiment, 24 rue Joseph Fourier, 38400 Saint Martin d’Hères, France; francois.gaudaire@cstb.fr (F.G.); Christophe.MARTINSONS@cstb.fr (C.M.); 6Division Eclairage et Champs Electromagnétiques, Centre Scientifique et Technique du Bâtiment, 11 rue Henri Picherit, 44300 Nantes, France; nicolas.noe@cstb.fr

**Keywords:** work, Wifi, radiofrequencies, measurements, libraries

## Abstract

Background and Objectives: With the spread of Wifi networks, safety concerns have arisen, with complaints of somatic disorders, notably in traditional libraries and media libraries. The aim of the present study was to describe the conditions and levels of exposure to radiofrequency electromagnetic fields in the real-life occupational conditions of those working in traditional libraries and media libraries. Methods: Dynamic measurements, using an exposimeter, were taken in 20 radiofrequency bands from 88 to 5850 MHz. The activity of 28 library workers was analyzed on a space-time budget. An audit of exposure sources and static measurements enabled the work-places to be mapped. Results: In seven libraries, 78,858 samples were taken over the 20 radiofrequency bands from 88 to 5850 MHz. Exposure was described for 28 working days. The median total field was 0.071 V/m (10th percentile: 0.022 V/m, 90th percentile: 0.534 V/m) and for Wifi the median field was 0.005 V/m (10th percentile: 0.005 V/m, 90th percentile: 0.028 V/m). Median individual exposure to Wifi frequency waves ranged from 0.005 to 0.040 V/m. Conclusions: Overall, the occupational exposure in this sector was close to the exposure in the general population. Peaks were due to the use of walkie-talkies by security staff. Exposure due to external sources depended on geographic location. Exposure in this occupation is well below the general occupational exposure levels, notably as regards Wifi.

## 1. Introduction

A description of electromagnetic field exposure (classified by the International Agency for Research on Cancer, IARC, as category 2B: i.e., possibly carcinogenic) is today a social demand both to guarantee system quality and to meet the concerns of public opinion regarding the multiplication of transmitters (base station antennae, Wifi, etc.) [1]. The uses of new technologies and the characteristics of their signals (modulations, frequency bands, intensity) impact individual exposure levels.

To be able to communicate and to inform the general public, it is important to have tools that quantify electromagnetic fields. A better knowledge of the typical exposure and a comparison between the general and the occupational population is important to assess a possible risk. Previous studies have evaluated the exposure in the general population, that it was inconstant according to the methodology and variable over the time [2,3,4]. There is an occupational dimension here: There are ever more requests for exposure assessments in order to evaluate occupational health risks and ensure appropriate monitoring of those exposed. Occupational health personnel often have no way of assessing the occupational radiofrequency exposure of those concerned, measurement methods being complex and not well known. Most occupational studies use fixed measurements to map work-stations and work-places, which provide only values at a given time-point without taking account of variation over the working day or subjects’ movements [5,6]. Some studies have used exposimeters, to be able to monitor employees at their work-stations; but close-field exposure is poorly assessed by this means [7]. Combining the two methods gives a better picture of exposure, and has been used in certain work environments [8].

With the spread of Wifi networks, safety concerns have arisen, with complaints of somatic disorders, notably in traditional libraries and media libraries: For example, in Paris in the summer of 2007, during the “*Nouvel élan pour Paris, ville numérique*” campaign. These concerns are taken up by the media, and employees demand details on exposure. Simply checking that norms are being respected, as proposed at an international level in 1998 by the ICNIRP (International Commission on Non-Ionizing Radiation Protection) failed to alleviate the worry, which especially concerned the existence and probability of non-thermal effects [9,10]. For these reasons, occupational physicians covering library workers asked us to study the issue.

Our objective was to describe radiofrequency electromagnetic field exposure conditions and levels in the real-life working conditions of traditional and media library employees, with a view to improving characterization and ultimately proposing a relevant form of monitoring.

## 2. Methodology

### 2.1. Site Characterization

There are 14 traditional or media libraries in the city of Lyon, France. Seven were selected, on the basis of their characteristics (size, traditional, or media, etc.), possible exposure sources (Wifi, automated lending terminals, etc.), types of work-station, and of study logistics (working hours of employees and of the two study teams involved in each site).

After site selection, data were collected from the relevant organizations and a study visit was paid to each site. This step enabled the collection of certain information.

1. Identification of indoor radiation sources: RFID (radiofrequency identification) systems for book, compact disc, and DVD lending; antitheft gates; DECT (digital enhanced cordless telecommunication) phones; cell-phones; Wifi bases; other sources detected in the environment and emitting significant electromagnetic fields.

2. Identification of outside sources: Radioelectric transmitters (cell-phone, broadcasting) identified from the French national radiofrequency agency (ANFR) data.

3. Two- or 3-D building plans, if available.

### 2.2. Employee Selection

Employees were volunteers, selected for their presence on-site on the day fixed measurements were made, taking account of the occupational profile (few agents on small sites). Twenty-eight persons were thus selected: In other words, 2–13 per site, depending on the size, or 5.7% of overall personnel.

### 2.3. Measurement Strategy and Equipment

#### 2.3.1. Measurement Strategies

The chosen methodology consisted in mapping the work-space after inventorying possible exposure sources, using static measurements and dynamic measurements, while taking account of employees’ activities during the working day.

Measurements were made by fixed detectors and by exposimeters, and were limited to radiofrequencies.

The measurement protocols enabled:rapid analysis of the whole site, to identify relevant areas: Spatial scanning, and configuration analysis of sources, work-stations, and places where personnel tend to stay;adaptation of measurement procedures according to frequency bands and equipment used.

#### 2.3.2. Materials

The procedures followed the ANFR measurement protocol (ANFR DR15-3).

(a) The SATIMO EME-SPY 200 exposimeter (Microwave Vision Group Industries Bretagne, 29280 Plouzané, France), covering 20 bands from 88 to 5850 MHz, was used as a portable device for prolonged measurements (see Table 1). Detection threshold for FM, TV3, TETRA, TV4 and 5 and Wifi 5G bands was 0.01 V/m, and 0.005 V/m for LTE 800, GSM, DCS, DECT, UMTS, Wifi 2G, LTE 2600 and WiMax. For all bands, the maximum detection limit was 6 V/m (95.49 mW/m^2^).

As the device had limited autonomy and it was not possible to change the batteries during the working day, measurement times were set at 8 s, covering the whole working day.

(b) Spectrum analyzer and isotropic probe (Narda SRM 3006 + accessories) for radiofrequencies. The analyzer was connected to a NARDA 420 MHz-6 GHz or 27 MHz-3 GHz antenna via a 50 Ohm coaxial cable, and allowed spectrum analyses and electromagnetic field measurements in RMS (root mean squared) values for the frequency bands in question.

(c) Narda EHP 200 probe for intermediate frequency measurements.

#### 2.3.3. Measurement Implementation

At each site, measurements were made simultaneously by fixed probes and by exposimeters worn by the participants. During a visit for the preparation of the measures, a plan of the workspaces was recovered with the localization, in particular, of the Wifi terminals. The static measurements were made on a course in the workspace selected on plan allowing a scan of it.

(a) Individual samples

Each employee was equipped, during 1 working day, with an exposimeter, worn on one side or the other at waist level, and also filled out a space-time budget questionnaire (STB) tracking his or her movements and activities in the library (telephoning, RFID use, etc.) during the measurement period. The exposimeter was recalibrated before each working session.

(b) Environmental exposure measurements

The measurement method used “imposed” targets and “permanent” electromagnetic radiation, such as Wifi networks or cell-phone base stations. It is based on the measurement protocol of the French Frequencies Agency (reference ANFR/DR15-3), which follows the European standard EN50,492, which is the basic norm for measuring on-site electromagnetic fields in regard to human body exposure in the vicinity of base stations.

An initial analysis was made at each site, to map transmitters and their principal irradiation areas. Transmitters were located on the various site plans. 

These data were collected visually, based on information gathered on-site and on a search of the ANFR data-base (www.cartoradio.fr), or by selective spectrum measurements. This step determined the frequency bands to be analyzed: For example, 100 MHz to 6 GHz.

At each site, measurement areas were more precisely determined on the basis of the expression of demand, transmitter topology, and irradiation areas.

In a second step, fixed positions in the environment were analyzed to determine the various services to be assessed. Measurements used the NARDA SRM-3006 spectrum analyzer. Total exposure was measured between 420 MHz and 6 GHz.

Measurements were then made along a defined pathway, to quantify the total exposure for an individual in the library. These measurements were made at a constant speed, at a height of 1.7 m, along a pathway based on site geometry and personnel movements within the library.

### 2.4. Data Analysis

Exposimetry results were analyzed globally and per individual, per site, and per frequency.

Working time was collected from the STBs filled out at the time of each measurement. As the exposimeter stopped only when the battery had run down, meal breaks and time spent off-site were excluded from final analyses.

Descriptive data were expressed as mean and median time spent above the device detection threshold (0.005 V/m or 0.01 V/m, depending on frequency), and the various thresholds were calculated.

### 2.5. Ethics, Consent, and Health-and-Safety Submission

The methodology was presented to the health and safety committee (CHSCT), as were results for each library (in December 2016).

## 3. Results

### 3.1. Individual Samples

Per frequency, 78,858 samples were analyzed: In other words, a mean 2846 per frequency per person. For each frequency, individual samples were taken every eight seconds over the working day: In other words, the total recording time, 175 h 14 min.

Kinetics, as expected, depended on the source in question and its functioning. Measurements comprised two categories: Indoor (notably, Wifi and DECT) and outdoor (base station downlink, and radio and broadcasting emissions).

(a) Indoor sources

Indoor sources mainly comprised DECT and 2G Wifi. For each frequency band, except for 5G Wifi, the median was equal to the detection threshold (Figure 1).

2G Wifi exposure above the 0.005 V/m detection threshold was found 34.06% of the time, and above 0.5 V/m 0.02% of the time (Table 2). Mean individual 2G Wifi exposure ranged from 0.005 V/m (detection threshold) to 0.042 V/m. Individual kinetics were relatively low-level, with little variation (range 10–90th percentile: 0.005–0.028 V/m) (Table 3). The maximum was 2.89 V/m, found at a single site, for a single employee whose mean level was fairly low, at 0.09 V/m, with 14.62% of measurements >0.005 V/m; in the same period, there were six peaks, ranging from 0.86 V/m to the maximum, corresponding to the use of a microwave oven by an employee, identified on the STB.

For 5G Wifi, the exposure was above the detection threshold of 0.01 V/m for 3.37% of overall work-time.

(b) External sources

Our measurements assessed exposure due to all transmitters in the environment: Radio, television, cell-phone network relays, etc. Downlink network emissions predominated in these “external” exposures (Figure 2). External exposure was highly site-dependent, with geographic and temporal specificities. FM radio transmitter exposure was above the exposimeter detection threshold in respectively 100, 98, and 86% of samples taken in the three city-center sites, and in less than 50% of the others; this matches transmitter locations in Lyon. Overall, above-threshold exposure (>0.01 V/m) for TV and radio transmitter frequencies concerned the FM band in 61.69% of cases, TV3 in 5.02%, and TV4 and 5 in 11.71% (Table 2); however, only 32 FM samples (0.04%), and no TV3 or 4 and 5 samples, exceeded 0.5 V/m. Maximum levels were 1.093 V/m for FM, 0.328 V/m for TV3 and 0.409 V/m for TV4 and 5.

PMR (private mobile radiocommunications) transmitters (mainly walkie-talkies) were taken into account in exposimetry. Significant levels were found in 2.31% of measurements for TETRA I and TETRA II, and in 0.82% for TETRA III, independently of site or work-station. The median level was 0.01 V/m. Over the whole working day, only TETRA III levels exceeded 5 V/m (with 16 measurements reaching the device detection ceiling of 6 V/m); this concerned two individuals, working in the security post, and was due to walkie-talkie use (Table 2). In the largest site, in a dense urban area with many transmitters in the vicinity, the external exposure varied from floor to floor and room to room (Figure 3).

Downlink cell-phone emissions included new frequencies, with 4G networks being deployed during and especially toward the end of the study period: LTE 2600 exposure was above the detection threshold in 49.06% of cases overall, and in 70.79% of measurements taken at the last site in June 2015.

For DECT (fixed cordless phones), mean per site ranged from 0.006 to 0.015 V/m and per individual from 0.0052 to 0.0337 V/m. The 90th percentile overall was 0.021 V/m.

(c) Comparison between external and internal sources for the total field

Median total field on exposimetry was 0.03 V/m (range 10–90th percentile: 0.019–0.131) from external sources and 0.025 V/m (0.02–0.063) from internal sources. Depending on the site, maximum total external field ranged from 0.31 to 2.21 V/m.

### 3.2. Static Pathway Measurements

Static measurements mapped exposure levels along work-place pathways. Maximum RMS in the site with the highest mean value was 1.95 V/m along the Wifi pathway; in sites liable to show the greatest exposure, nearest to sources and work-stations, maximum RMS was 0.15 V/m.

Figure 4a,b showed results of the samples made on a pathway in a typical area of a public library (La Part Dieu—Lyon). The starting point of the pathway was at the desk of the library section. The maximum RMS value of 4.5 V/m, observed on the window side, correspondeds mainly to UMTS downlink at 2100 MHz with 4.337 V/m. Close to the desk, the observed field corresponded to the Wifi 2400 MHz and GSM 900 MHz with a total value of 2.1 V/m. The Wifi spots were at a height of 2 m. These measurements corresponded to the maximum values observed from static exposure in all investigated libraries.

This initial on-site study also gave an overview of a particular exposure situation, with lending terminals and antitheft gates, investigated by another method (exposimeter does not permit to explore this frequency) in a complementary study not reported here.

## 4. Discussion

The originality of the present study was to assess radiofrequency exposure in library workers during the working day on dynamic and static measurements, in parallel to a study of tasks performed.

Pathway measurements provided an overall scan of the work-space, with maximum exposure values, and mapped exposure in each work-place.

Individual exposimetry determined instantaneous values throughout the working day, and thus peaks corresponding to tasks, places, and time of day.

Although this was not an initial study objective, the results showed that exposure, notably to Wifi transmitters, was well below ICNIRP limits, whatever the frequency.

Motivating this study, the big fear in the profession was exposure to Wifi, as it is being massively deployed in traditional and media libraries.

Mean individual Wifi exposure exceeded the median, which equaled the device’s detection threshold. Only five measurements exceeded 1 V/m, corresponding, in fact, to an on-site microwave oven for meal breaks: The emission frequency did not differ, but including the microwave oven in the space-time budget enabled exposure due to these two devices to be distinguished. On static measurement, the maximum threshold was 0.9 V/m (2.15 mW/m^2^), comparable to results in Australian schools equipped with Wifi [11]. Mean Wifi levels overall were lower than for radio and cell-phone network emissions, whereas in the Australian study they were similar inside classrooms and lower outdoors [11].

Mean Wifi exposure was comparable between sites, except for one with a slightly higher level (0.04 V/m); this was a medium-size library, with two employees both showing comparable exposure; mapping on static measurement found an RMS level of 1.95 V/m (10.09 mW/m^2^). The total field on exposimetry for all sites was comparable to those reported elsewhere for possibly similar work-places. Van Wel, studying classrooms in Amsterdam equipped with Wifi, reported a mean value slightly lower than in the present study: That of 0.16 V/m (0.07 mW/m^2^) versus 0.19 V/m (0.1 mW/m^2^ [12]. Bolte, in the general population, on the other hand, reported a higher value: That of 0.260 V/m (0.18 mW/m^2^) [13]. In these studies, the highest values were associated with cell-phones and DECT.

Van Wel’s protocol, however, had certain differences: The total field was calculated over fewer bands, as an earlier model of the exposimeter was used, and was probably underestimated. Exposimeters were moreover used for static measurements lasting two minutes, to map classrooms; measurements were made after class and cell-phones were turned off or kept at a distance, and thus not taken into account. The protocol was suitable for mapping site exposure, but to the exclusion of human activity.

The present overall exposure results were comparable to those of a previous general population study also conducted in Lyon—0.201 V/m in 2009; Wifi exposure was low, with only 14% of measurements above the 0.05 V/m detection threshold, 95% of which were between 0.06 and 0.1 V/m; the number of above-threshold measurements was greater for Wifi than cell-phones (fewer than 5% > 0.05 V/m, but with 95% confidence intervals of 0.06–3.37 V/m for the GSM band and 0.05% > 10 V/m). In the present study, exposure due to the downlink bands was higher and exposure due to uplink emissions lower [14]. As in Chiaramello’s review, the indoor exposure depends on both outdoor and indoor environments [15]. The maximum mean exposure in “offices” and in “other public buildings” is more than our value [15].

The present study concerned occupational exposure. In Bolte’s study, the total field during the working day (0.305 V/m, 0.25 mW/m^2^) was independent of the particular occupation [13].

Some occupational sectors involve high radiofrequency exposure: Telecommunications, metallurgy (industrial microwave ovens, arc welding, magnetoscopic testing (^1.^ Non-destructive test method, widely used in materials technology), dielectric heating, high-frequency presses), or medicine (MRI, electric knife surgery, esthetic dermatology) [16]. Elsewhere, in the service sector and office work in general, exposure levels are comparable to those in the general population, but with a larger number of sources and more frequent use of computers, Wifi, RFID chips, alarms, remote controls, cell-phones and DECT phones.

Likewise, in the present study, library exposure levels were low, close to those in the general population [6,17,18]. Indeed, library workers are not counted as being exposed in occupation/exposure matrices [19,20].

### Study Strengths and Weaknesses

The interest of the exposimeter in the present protocol was that it could be worn throughout the working day, tracking real working conditions, including the employee’s movements and presence in the library of members of the public who might create exposure by using their cell-phone. The exposimeter enabled both a global approach and analysis in terms of time and frequency. It would have been interesting to analyze variations in exposure according to the working day: With or without members of the public, and numbers of persons using the Wifi network; this information was not available in the present study. For example, it was not possible to take account of whether the personnel’s computers were in sleep or active mode, which can affect the global level via the number of peaks but with little impact on maximum levels [11].

Whether exposimetric measurements are representative of real exposure is a question which arises especially in terms of measurement bias [7]. The devices were calibrated ahead of use; uncertainty of measurement for the model used (EME Spy 200) is reported to be 0.1–2.2 dB (vertical polarization) and 0.6–2.8 dB (horizontal polarization). Error may be induced by device positioning with respect to the body, due to waves being reflected by the body [21]. Blas et al. therefore recommended using two exposimeters, but this is too demanding for an entire working day. However, in the present study, exposimetric measurements could be confirmed by complementary static measurements.

A common criticism of exposimeters is that they cannot precisely measure close fields (when the source is close to the device) [7]. Exposimetry assesses the distant field, considered homogeneous over the whole body; but for close fields the approach is unsuited [22]. In the case of libraries this concerns measuring nearby internal sources, which certainly contribute to the total received field. Employees work close to emission sources, and close-field exposure has to be assessed observationally and by the space-time budget. Close internal sources especially comprise DECT phones, the individual’s cell-phone and Wifi bases. In practice, the parallel work-place study showed that other internal sources (Wifi base usually at least 1 m distant) were well described. DECT analysis was more difficult, as STBs did not mention DECT use associated with the main exposures at this frequency. On the other hand, when DECT use was recorded, it did not systematically correspond to an exposure peak. However, in the majority of the cases (except for DECT), the space-time budget is very informative and reported information correlates very well with the profile of exposure of each subject, centrally to other papers which estimate that the space-time budget poorly represent the activity of a person [13]. 

Another limitation of exposimeters lies in band selectivity, due to possible couplings between bands, explaining for example, the results on DECT.

The exposimeter (often wrongly referred to as a dosimeter) only measures exposure on the body surface; extrapolation to the dose received by the body and per organ would be of interest.

Finally, interpretation of the results has to take account of the technical limitations of the device, recording exposure levels between 0.005 and 6 V/m. Thus, analysis should be based on mean values. In the present case, where most values were close to the detection threshold, exposure was likely overestimated: Below-threshold fields were assimilated to the threshold value of 0.005 V/m. This threshold is very low and within the limits of uncertainty of measurement.

## 5. Conclusions

The aim of this study was to assess electromagnetic exposure of employees in libraries, notably, to Wifi.

On average, occupational exposure in this branch is close to that of the general population. The highest peaks were associated with walkie-talkie use by security staff. Exposure to outside sources depended on the geographic location. Employees’ exposure was largely below the occupational norms, notably as regards Wifi. Static exposure from radiofrequency outdoor emissions (UMTS, GSTM) and from indoor Wifi spots is found to be 10 times lower than electromagnetic limit values defined by ICNIRP.

The exposure of workers to electromagnetic fields is rarely explored by occupational practitioners due to a lack of methodology to perform this evaluation. The present methodology for describing exposure could be transposed to other occupational radiofrequency exposure contexts, given persisting doubts about the health risks involved and the current legislation.

### What is New in This Paper?

Exposure to radiofrequency electromagnetic fields due to new technologies in the real-life occupational conditions in library and media libraries is not well characterized. This study finds an exposure close to the general population. However, high peaks were due to the use of walkie-talkies by security staff. Exposure due to external sources depended on geographic location.

## Figures and Tables

**Figure 1 ijerph-16-02087-f001:**
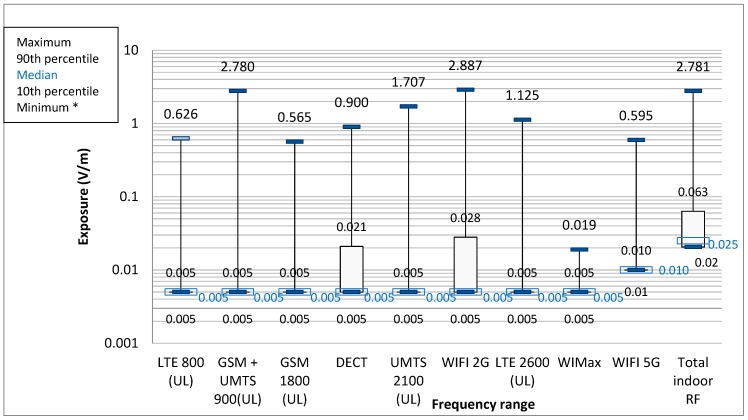
Exposure due to indoor sources. * Minimum is equal to the detection threshold of 0.01 V/m for the Wifi, and 0.005 V/m for other frequencies. The value of the minimum is not specified in the graph.

**Figure 2 ijerph-16-02087-f002:**
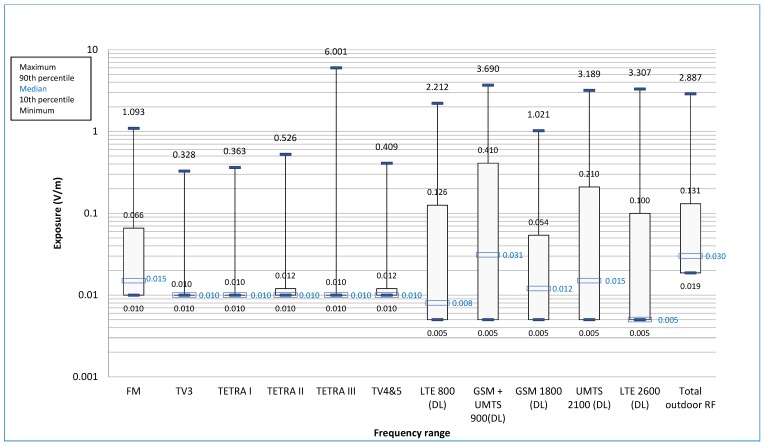
Exposure due to outdoor sources. The minimum value is equal to the detection threshold of 0.01 V/m for frequencies between FM and TV4 and 5, and 0.005 V/m for other frequencies. The value of the minimum is not specified in the graph. For all frequencies it is confused with the 10th percentile.

**Figure 3 ijerph-16-02087-f003:**
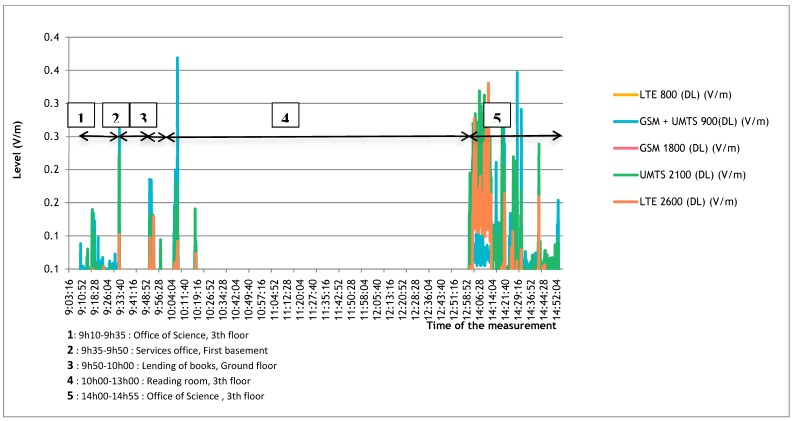
Example of individual exposure samples according to the geographical situation and the workplace.

**Figure 4 ijerph-16-02087-f004:**
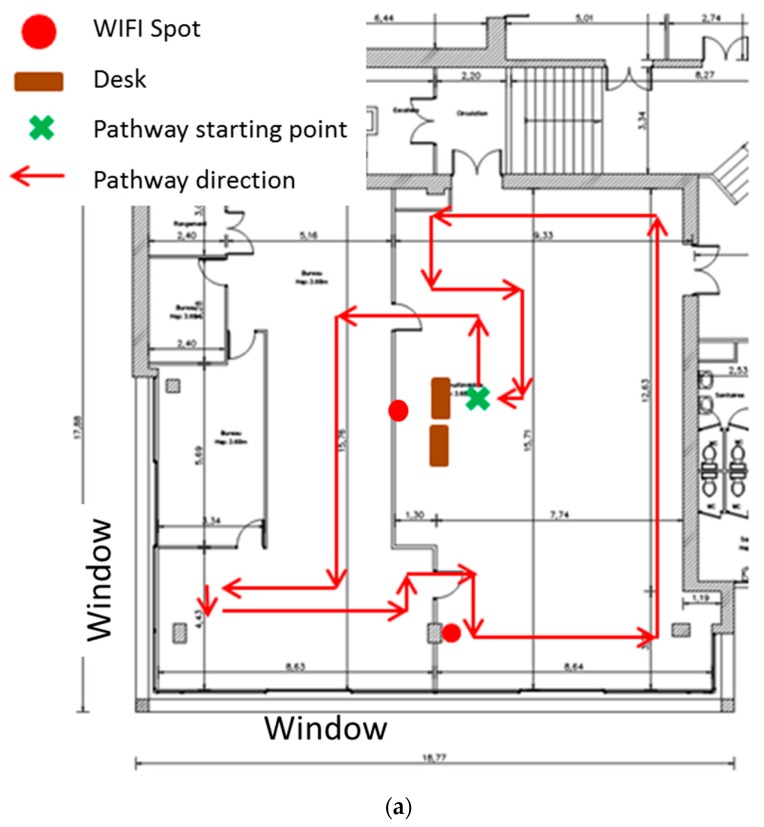
(**a**) Static exposure along a pathway: Example of electromagnetic field field measurement pathway—Library Part-Dieu, Level 3 (digital documentation area); (**b**) example of electric field measurement results on a pathway—Library Part-Dieu, Level 3 (digital documentation area).

**Table 1 ijerph-16-02087-t001:** Frequency ranges explored by the SATIMO EME-SPY 200 exposimeter.

Frequency Ranges	Frequency Min (MHz)	Frequency Max (MHz)
Broadcast network FM	87	107
Broadcast network TV3	174	223
Cellular network TETRA I	380	400
Cellular network TETRA II	410	430
Cellular network TETRA III	450	470
Broadcast network TV 4 & 5	470	770
Cellular network LTE 800 (DL ^1^)	791	821
Cellular network LTE 800 (UL ^2^)	832	862
Cellular network GSM + UMTS 900 (UL ^2^)	880	915
Cellular network GSM + UMTS 900 (DL ^1^)	925	960
Cellular network GSM 1800 (UL ^2^)	1710	1785
Cellular network GSM 1800 (DL ^1^)	1805	1880
DECT (Cordless phones)	1880	1900
Cellular network UMTS 2100 (UL ^2^)	1920	1980
Cellular network UMTS 2100 (DL ^1^)	2110	2170
Wifi 2G	2400	2483.5
Cellular network LTE 2600 [UL ^2^)	2500	2570
Cellular network LTE 2600 (DL ^1^)	2620	2690
WIMax	3300	3900
Wifi 5G	5150	5850

^1^ DL: Down link; ^2^ UL: Up link.

**Table 2 ijerph-16-02087-t002:** Results of 78,858 measurements per frequency (every eight seconds) during all the working time of all employees at all sites.

Frequency	Mean (V/m)	Standard Deviation (V/m)	Percentage of Samples > Detection Threshold in V/m	Percentage of Samples > 0.5 V/m	Number of Samples > 5 V/m
FM ^a^	0.031	0.045	61.69%	0.04%	0
TV3 ^a^	0.010	0.003	5.02%	0.00%	0
TETRA I ^a^	0.010	0.003	2.31%	0.00%	0
TETRA II ^a^	0.011	0.006	11.37%	0.00%	0
TETRA III ^a^	0.012	0.098	0.82%	0.06%	16
TV4&5 ^a^	0.012	0.008	11.71%	0.00%	0
LTE 800 (DL) ^b^	0.042	0.076	60.11%	0.41%	0
LTE 800 (UL) ^b^	0.005	0.004	1.12%	0.00%	0
GSM + UMTS 900 (UL) ^b^	0.007	0.034	4.58%	0.07%	0
GSM + UMTS 900 (DL) ^b^	0.122	0.202	89.80%	6.61%	0
GSM 1800 (UL) ^b^	0.006	0.005	9.68%	0.00%	0
GSM 1800 (DL) ^b^	0.025	0.039	78.05%	0.06%	0
DECT ^b^	0.010	0.011	39.70%	0.00%	0
UMTS 2100 (UL) ^b^	0.006	0.012	6.43%	0.01%	0
UMTS 2100 (DL) ^b^	0.072	0.140	73.23%	1.48%	0
WIFI 2G ^b^	0.011	0.024	34.06%	0.02%	0
LTE 2600 (UL) ^b^	0.006	0.009	3.15%	0.01%	0
LTE 2600 (DL) ^b^	0.035	0.080	49.06%	0.32%	0
WIMax ^b^	0.005	0.000	0.19%	0.00%	0
WIFI 5G ^a^	0.010	0.006	3.37%	0.00%	0
Total	0.190	0.278	99.08%	11.31%	16

^a^ detection threshold: 0.01 V/m. ^b^ detection threshold: 0.005 V/m.

**Table 3 ijerph-16-02087-t003:** Individual exposure samples for three frequency bands and total electromagnetic field.

	Site	Number of Measurements	GSM + UMTS 900 (UL)				GSM + UMTS 900 (DL)				Wifi 2G ^a^			CHAMP TOTAL ^b^				
			Mean	Standard Deviation	Maximum	Percentage of Measurements >0.005 V/m	Mean	Standard Deviation	Maximum	Percentage of Measurements >0.005 V/m	Mean	Standard Deviation	Maximum	Percentage of Measurements >0.005 V/m	Mean	Standard Deviation	Maximum	Percentage of Measurements >0.005 V/m
1	A	3628	0.01	0.01	0.23	2.40%	0.04	0.04	0.87	98.93%	0.008	0.008	0.092	30.13%	0.15	0.10	1.06	99.83%
2	A	2954	0.01	0.03	1.69	1.32%	0.04	0.03	0.40	99.83%	0.006	0.004	0.069	12.29%	0.12	0.11	1.74	99.93%
3	A	3199	0.01	0.09	2.67	11.88%	0.30	0.19	1.17	99.94%	0.005	0.002	0.056	1.81%	0.47	0.23	2.75	100.00%
4	B	2897	0.03	0.12	2.78	18.64%	0.01	0.00	0.05	21.88%	0.007	0.006	0.118	25.16%	0.04	0.12	2.78	95.65%
5	B	3293	0.02	0.05	1.16	20.07%	0.01	0.00	0.04	60.83%	0.007	0.007	0.135	41.66%	0.04	0.05	1.17	99.70%
6	C	2983	0.01	0.00	0.11	0.57%	0.03	0.02	0.18	99.70%	0.009	0.080	2.887	14.62%	0.06	0.09	2.89	99.90%
7	C	2749	0.01	0.00	0.07	0.87%	0.01	0.01	0.07	92.32%	0.006	0.002	0.055	27.21%	0.04	0.03	0.78	99.96%
8	D	2940	0.01	0.00	0.11	0.68%	0.01	0.02	0.33	80.85%	0.005	0.004	0.156	6.33%	0.13	0.08	1.12	100.00%
9	D	2541	0.01	0.00	0.21	0.55%	0.01	0.01	0.05	27.98%	0.005	0.001	0.035	1.42%	0.05	0.03	0.22	100.00%
10	E	3136	0.01	0.00	0.09	5.01%	0.08	0.08	0.63	100.00%	0.012	0.027	0.627	45.41%	0.14	0.17	1.33	100.00%
11	E	2586	0.01	0.00	0.08	8.51%	0.06	0.05	0.51	99.96%	0.012	0.010	0.116	58.97%	0.10	0.11	1.46	99.96%
12	F	1875	0.01	0.01	0.25	0.69%	0.01	0.00	0.02	80.59%	0.040	0.054	0.733	92.21%	0.05	0.06	0.73	97.49%
13	F	3561	0.01	0.02	0.80	10.50%	0.02	0.01	0.03	99.72%	0.042	0.013	0.248	99.97%	0.05	0.03	0.80	100.00%
14	C	2987	0.01	0.00	0.08	1.41%	0.01	0.01	0.07	99.36%	0.006	0.002	0.027	36.26%	0.05	0.01	0.29	100.00%
15	C	3010	0.01	0.01	0.33	2.52%	0.03	0.02	0.16	99.90%	0.005	0.002	0.051	12.82%	0.04	0.03	0.50	100.00%
16	G	2196	0.01	0.01	0.18	2.55%	0.21	0.35	3.69	97.81%	0.006	0.004	0.074	15.89%	0.37	0.66	4.14	98.95%
17	G	3422	0.01	0.02	1.17	2.63%	0.10	0.10	1.50	99.27%	0.013	0.014	0.156	57.39%	0.17	0.29	6.00	99.65%
18	G	2974	0.01	0.00	0.04	1.51%	0.17	0.17	1.72	99.56%	0.009	0.009	0.141	31.17%	0.23	0.22	2.71	99.73%
19	G	3333	0.01	0.00	0.08	0.66%	0.37	0.19	1.20	99.67%	0.011	0.015	0.119	19.32%	0.45	0.20	1.60	99.73%
20	G	2999	0.01	0.00	0.07	0.30%	0.53	0.27	2.51	99.90%	0.010	0.012	0.071	23.14%	0.63	0.29	2.58	100.00%
21	G	2118	0.01	0.00	0.01	0.09%	0.30	0.18	1.45	99.86%	0.006	0.012	0.490	1.84%	0.39	0.20	1.67	99.91%
22	G	2295	0.01	0.00	0.01	0.13%	0.42	0.26	3.26	100.00%	0.005	0.003	0.061	6.54%	0.61	0.34	3.52	100.00%
23	G	2038	0.01	0.01	0.50	0.69%	0.47	0.27	2.21	100.00%	0.017	0.020	0.150	35.62%	0.55	0.29	2.51	100.00%
24	G	1320	0.01	0.01	0.39	21.44%	0.02	0.02	0.17	70.30%	0.009	0.011	0.137	17.65%	0.07	0.13	1.91	98.26%
25	G	3625	0.01	0.01	0.20	4.08%	0.08	0.11	1.33	98.98%	0.016	0.019	0.542	71.86%	0.17	0.40	6.01	99.75%
26	G	2725	0.01	0.00	0.06	0.48%	0.16	0.18	1.48	100.00%	0.027	0.043	0.398	48.07%	0.22	0.19	1.57	100.00%
27	G	2616	0.01	0.06	1.72	8.94%	0.01	0.01	0.16	77.83%	0.007	0.006	0.104	19.57%	0.03	0.06	1.72	84.21%
28	G	2858	0.01	0.01	0.28	1.05%	0.02	0.02	0.22	88.73%	0.010	0.008	0.090	69.10%	0.04	0.03	0.54	98.95%

^a^ Results returned with three digits to avoid rounding at the same value; ^b^ Total field measured by the exposimeter.

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
