# Peer review of "Assessment of Personal Occupational Exposure to Radiofrequency Electromagnetic Fields in Libraries and Media Libraries, Using Calibrated On-Body Exposimeters"

_ijerph, 2019, doi:10.3390/ijerph16122087_

Round 1

Reviewer 1 Report

The manuscript presents results of interesting studies on the important problem of the characteristic of personal exposure to radiofrequency electromagnetic radiation in libraries using WiFi networks. The study was well done, but description of the results needs improvement. It is also advised to consider redrafting of English terminology used in the manuscript. Please consider more detailed comments and modifications of the manuscript with respect to the following:

- in the manuscript the results are mentioned as the “exposimeter measurements” – please consider to use usual wording “samples” recorded by exposimeters and specify is sample is counted in each frequency band separately or is all frequency bands

- description of results with the use of standard deviation is not correct when SD values exceed mean values (indicating that the distribution of analyzed set of data is not normal) – please consider to use the most universal statistical parameters – median and percentiles

- over last years very many papers have been published on the results of personal exposimetric measurements – please consider broader discussion of published data in the introduction and discussion sections

- page 2, line 18 – “infra-thermal health impact” is mentioned – please consider the modification of wording – have you mentioned there “non-thermal effects” of exposure?

- the level of exposure to electromagnetic radiation emitted by WiFi facilities is varying in time when the number of users of wireless network is changing; it is also very localize exposure in the vicinity of the emitting antennas – please try to characterize more precisely the scale of use of WIFI networks in the locations where reported measurements were done, as well as the location of emitting antennas with respect to the locations accessible to workers during exposimetric measurements – such information is important from the point of view of comparison of reported results with exposure situations in other locations

- figure 2 – please change “Champ total” to English

- table 1 – the use of SD parameter seems to be not justified (SD > aver in many cases); results of measurements in TETRA III frequency band (16 samples exceeding 5 V/m) need to be discussed in details

  table 2 – the use of SD parameter seems to be not justified; please explain more detailed exposure from the microwave oven mentioned at page 11

- discussion section – please unify references to power density – mW/m2 seems to be practical

- please provide more detailed info regarding the mentioned RFID facilities – it is not clear if exposures from it was covered by exposimetric or spot measurements – more detailed information from spot measurements is also welcome with respect to various sources mentioned in the discussion section

- at page 15 – lines 40-45 – some problems with interpretation of results of measurements recorded in the DECT frequency band are mentioned – please consider the possibility that DECT recordings may be a result of so-called out-of-band recording of radiation from different sources

- the first sentence of conclusion seems to be not linked with a results of measurements discussed in this paper – please delete it or provide more detailed description of the results to justify it.

- the antitheft gates mentioned in the conclusions also seems to be not linked with a results discussed in this paper – please provide more detailed description of this kind of exposure and if it was covered by exposimetric measurements or delete it.

- please consider re-drafting of the last sentence of conclusions to explain more precisely relation to mentioned “clinical monitoring”.

Author Response

The manuscript presents results of interesting studies on the important problem of the characteristic of personal exposure to radiofrequency electromagnetic radiation in libraries using WiFi networks. The study was well done, but description of the results needs improvement. It is also advised to consider redrafting of English terminology used in the manuscript. Please consider more detailed comments and modifications of the manuscript with respect to the following:

- in the manuscript the results are mentioned as the “exposimeter measurements” – please consider to use usual wording “samples” recorded by exposimeters and specify if sample is counted in each frequency band separately or is all frequency bands :

Response: we replaced “Measurements “ by “samples”: For each subject, sample was counted for each frequency band separately

- description of results with the use of standard deviation is not correct when SD values exceed mean values (indicating that the distribution of analyzed set of data is not normal) – please consider to use the most universal statistical parameters – median and percentiles.

Response: OK; we agree with the reviewer median and percentiles are more accurate : we replaced mean and SD by median and percentiles in the figures. The difficulty is that often results in papers are presented as mean, so we had chosen to present also mean and SD for comparisons in table 1.

- over last years very many papers have been published on the results of personal exposimetric measurements – please consider broader discussion of published data in the introduction and discussion sections

Response: we completed the introduction and the discussion

- page 2, line 18 – “infra-thermal health impact” is mentioned – please consider the modification of wording – have you mentioned there “non-thermal effects” of exposure?

Response: Yes, we corrected

- the level of exposure to electromagnetic radiation emitted by WiFi facilities is varying in time when the number of users of wireless network is changing; it is also very localize exposure in the vicinity of the emitting antennas – please try to characterize more precisely the scale of use of WIFI networks in the locations where reported measurements were done, as well as the location of emitting antennas with respect to the locations accessible to workers during exposimetric measurements – such information is important from the point of view of comparison of reported results with exposure situations in other locations

Response: As we explained each worker use a Space-Time Budget questionnaire. It was possible to locate the workers during the measurement time (in comparison with a plan of the workplace) and the use of Wifi equipments.  We completed the methodology with the following sentenceDuring a visit of preparation of the measures, a plan of the workspaces was recovered with the localization in particular of the terminals Wifi. The static measurements were made on a course in the workspace selected on plan allowing a scan of it.”

- figure 2 – please change “Champ total” to English

Response: done

- table 1 – the use of SD parameter seems to be not justified (SD > aver in many cases); results of measurements in TETRA III frequency band (16 samples exceeding 5 V/m) need to be discussed in details

Response: Over the whole working day, only TETRA III levels exceeded 5V/m (with 16 measurements reaching the device detection ceiling of 6V/m); this concerned 2 individuals, working in the security post, and was due to walkie-talkies use. See page 10, lines 16-18

  table 2 – the use of SD parameter seems to be not justified; please explain more detailed exposure from the microwave oven mentioned at page 11

Response: as requested by the reviewer above, we replaced the mean and SD by the median and ranges between the 10th and the 90TH percentiles . In the table we let the mean and the SD because often results in papers are presented as mean, so we had chosen to present also mean and SD for comparisons in table 1 and 2.

In one library, employees could use a microwave oven in a room (as it was mentioned in the STB), situated at the level of the work rooms, where they can warm some dishes at noon. This point is discussed page 15, lines 18-21

- discussion section – please unify references to power density – mW/m2 seems to be practical

Response: we unified references to power density (mW/m2), also we kept V/m because a majority of papers gave their results in this unit.

- please provide more detailed info regarding the mentioned RFID facilities – it is not clear if exposures from it was covered by exposimetric or spot measurements – more detailed information from spot measurements is also welcome with respect to various sources mentioned in the discussion section

Response: the RFID facilities were explored through another methodological pathway. We are writing a specific paper . It is the reason why we say in the results part that this will be report in a complementary paper (page 15, lines 1-3. We deleted the sentence corresponding to this part in the method section: page 5 lines  7-8.

·         Finally, specific close-field measurements were made in the laboratory on specific sources, such as RFID terminals. These results are not presented here.

- at page 15 – lines 40-45 – some problems with interpretation of results of measurements recorded in the DECT frequency band are mentioned – please consider the possibility that DECT recordings may be a result of so-called out-of-band recording of radiation from different sources

Response: we completed the discussion on this point; see page 16 lines 42-43: “Another limitation of exposimeters lies in band selectivity, due to possible couplings between bands, explaining for example, the results on DECT.”

- the first sentence of conclusion seems to be not linked with a results of measurements discussed in this paper – please delete it or provide more detailed description of the results to justify it.

Response: We agree; we deleted the first sentence and changed somewhat the conclusion.

- the antitheft gates mentioned in the conclusions also seems to be not linked with a results discussed in this paper – please provide more detailed description of this kind of exposure and if it was covered by exposimetric measurements or delete it.

Response: the reviewer is right, the place of this sentence should not be in the conclusion ; this  sentence was placed at the end of the results part, in order to signal to the reader that RFID and antiheft gates represent a specific problem, which will be explored elsewhere.

- please consider re-drafting of the last sentence of conclusions to explain more precisely relation to mentioned “clinical monitoring”.

Response: we completed this sentence by another one

“The exposure of workers to EMF are rarely explored by occupational practitioners due to a lack of methodology to perform this evaluation. The present methodology for describing exposure could be transposed to other……..”

Reviewer 2 Report

In this paper the authors assessed RF-EMF exposure in libraries measured by both personal exposimeters and spot measurements.

I have the following concerns/suggestions for the authors:

As in the last years a huge research effort was done towards the assessement of RF-EMF exposure in indoor public places (see, e.g., Sagar, S,  et al,  J. Expo. Sci. Environ. Epidemiol, 2018; Urbinello, D.et al, Environ. Res, 2014; Chiaramello, E, et al, Int. J. Environ. Res. Public Health 2019, Foster, K.et al, Health Phys. 2013): I suggest the authors to better substantiate the state of art in the "Introduction" section and to better discuss their results.  

The authors should add the description (maybe in a table) of the frequency bands in which the exposure was measured, as now the manuscript presents just the technologies to which each frequency band refers.

Results section: results are presented in an unclear way (e.g. figures 1 and 2 shows values about indoor and outdoor sources, but the text describes the results in the opposite order).

It is not clear to me which is the difference between fig.1 and 2 and Table 1: does this latter just add (compared to the figures) information about non-detect values? I would suggest the authors to better present results, to make easier for the reader to understand what is presented and avoid redundancy.

Very poor results are shown for the spot measurements: maybe the authors could add some information and discuss their findings also for this type of procedure.    

Author Response

I have the following concerns/suggestions for the authors:

As in the last years a huge research effort was done towards the assessement of RF-EMF exposure in indoor public places (see, e.g., Sagar, S,  et al,  J. Expo. Sci. Environ. Epidemiol, 2018; Urbinello, D.et al, Environ. Res, 2014; Chiaramello, E, et al, Int. J. Environ. Res. Public Health 2019, Foster, K.et al, Health Phys. 2013): I suggest the authors to better substantiate the state of art in the "Introduction" section and to better discuss their results.  

Response: we completed the introduction and the discussion, and added references to these publications

The authors should add the description (maybe in a table) of the frequency bands in which the exposure was measured, as now the manuscript presents just the technologies to which each frequency band refers.

Response: we added such a table in the methodology part of the paper

Results section: results are presented in an unclear way (e.g. figures 1 and 2 shows values about indoor and outdoor sources, but the text describes the results in the opposite order).

Response: As requested by the reviewer, we changed the order  of the presentation in the text of the indoor and outdoor results

It is not clear to me which is the difference between fig.1 and 2 and Table 1: does this latter just add (compared to the figures) information about non-detect values? I would suggest the authors to better present results, to make easier for the reader to understand what is presented and avoid redundancy.

Response: In the figures we presented the median and the 10th  and 90th percentiles of the outdoor and indoor exposures and the total field. The table 2 presented the total exposure for each band and the percentage of samples above the detection threshold, 0.5 and 5 V/m

Very poor results are shown for the spot measurements: maybe the authors could add some information and discuss their findings also for this type of procedure.    

 Response: we added  figures 4a and 4b which give an example of  the strategy of measurements and the corresponding results for the spot measurements. We completed the results with the following sentences:

“Static measurements mapped exposure levels along work-place pathways. Maximum RMS in the site with the highest mean value was 1.95V/m along the WiFi pathway; in sites liable to show the greatest exposure, nearest to sources and work-stations, maximum RMS was 0.15V/m.

Figures 4a and 4b showed results of the samples made on a pathway in a typical area of a public library (La Part Dieu - Lyon). The starting point of the pathway is at the desk of the library section, a maximum RMS (root mean square) value of 4.5V was observed on the window side, corresponding mainly to UMTS downlink at 2100MHz with 4.337 V/m. Close to the desk, the observed field corresponds to the Wifi 2400MHz and GSM 900MHz with a total value of 2.1V/m. The Wifi spots are at a height of 2m. These measurements corresponded to the maximum values observed from static exposure in all investigated libraries.

This initial on-site study also gave an overview of a particular exposure situation, with lending terminals and antitheft gates, investigated by another method (exposimeter doesn’t permit to explore this frequency) in a complementary study not reported here.”

Round 2

Reviewer 2 Report

The authors answered all my comments/suggestions.